

# A quantitative approach on environment-food nexus: integrated modeling and indices for cumulative impact assessment of farm management practices

Shervin Jamshidi and Anahita Naderi

Department of Civil Engineering, University of Isfahan, Isfahan, Iran

## ABSTRACT

**Background:** Best management practices (BMPs) are promising solutions that can partially control pollution discharged from farmlands. These strategies, like fertilizer reduction and using filter strips, mainly control nutrient (N and P) pollution loads in basins. However, they have secondary impacts on nutrition production and ecosystem. This study develops a method to evaluate the cumulative environmental impacts of BMPs. It also introduces and calculates food's environmental footprint (FEF) for accounting the total environmental damages per nutrition production.

**Methods:** This study combines the soil and water assessment tool (SWAT) for basin simulation with the indices of ReCiPe, a life cycle impact assessment (LCIA) method. By these means, the effectiveness of BMPs on pollution loads, production yields, and water footprints (WFs) are evaluated and converted as equivalent environmental damages. This method was verified in Zrebar Lake, western Iran. Here, water consumption, as WFs, and eutrophication are the main indices that are converted into equivalent health and ecological impairments. Two methods, entropy and environmental performance index (EPI), are used for weighting normalized endpoints in last step.

**Results:** Results showed that using 25–50% less fertilizer and water for irrigation combined with vegetated filter strips reduce N and P pollution about 34–60% and 8–21%, respectively. These can decrease ecosystem damages by 5–9% and health risks by 7–14%. Here, freshwater eutrophication is a more critical damage in ecosystem. However, using less fertilizer adversely reduces total nutrition production by 1.7–3.7%. It means that BMPs can decline total ecological damages and health risks, which threatens nutrition production. FEF presents a tool to solve this dilemma about the sustainability of BMPs. In the study area, a 4–9% decrease in FEF means that BMPs are more environmental friendly than nutrition menacing. Finally, this study concludes that SWAT-ReCiPe with FEF provides a quantitative framework for environment-food nexus assessment. However, due to the uncertainties, this method is recommended as a tool for comparing management strategies instead of reporting certain values.

Corresponding author
Shervin Jamshidi,
sh.jamshidi@eng.ui.ac.ir

# INTRODUCTION

Best management practices (BMPs) are promising solutions for controlling pollution discharged from non-point sources (NPS), including agriculture (*Liu et al., 2017*). Phosphorous (P) and nitrogen (N) compounds are typical pollutants transported in basins from farmlands (*Hanief & Laursen, 2019*). Water quality degradation and eutrophication are the possible results of these emissions. Filter strips (FS) (*Merriman et al., 2019*), fertilizer reduction (FR) (*Geng, Yin & Sharpley, 2019*), no-tillage farming (*Plunge, Gudas & Povilaitis, 2022*), tracing and fencing (*Sheshukov et al., 2016*), constructed wetlands (*Li et al., 2021b*), straw mulching (*Jang, Ahn & Kim, 2017*), or changing crop patterns and land-uses (LUs) (*Plunge, Gudas & Povilaitis, 2022*) are some examples of BMPs. Although these strategies have positive regional impacts on pollution transport (*Stubbs, 2016*), they may affect other ecosystems (*Čuček, Klemeš & Kravanja, 2015*), farmers' income (*Imani et al., 2017*), and even nutrition production. Therefore, assessing the effectiveness of BMPs needs detailed studies in basin scale with combined methods.

Some research has recently evaluated the effectiveness of BMPs. In the Great Lakes, *Merriman et al. (2019)* concluded that multiple BMPs combined with FS can reduce nutrients and sediment more significantly than single BMPs. Here, total phosphorus (TP) and total nitrogen (TN) removal could reach about 20% (*Merriman et al., 2019*). The results of *Liu et al. (2019)* similarly showed that combined BMPs with FS are more effective on reducing pollution load than individual BMPs. They recommended that modeling tools for cost-effective analysis can create a more sustainable framework for water quality improvement in agricultural basins (*Liu et al., 2019*). *Imani, Delavar & Niksokhan (2019)* also recommended to set priorities for BMPs in critical areas according to their TN and TP reduction and related costs (*Imani, Delavar & Niksokhan, 2019*). Modeling with field surveys verified that BMPs can reduce 25% nutrient pollution in a basin while sediment entrapment in the riparian zone can develop organic nutrient removal to about 60%. FS can solely act as an effective BMP with 20% TP removal (*Sheshukov et al., 2016*). Nonetheless, BMPs may reduce the runoff and adversely concentrate pollutants downstream (*Jang, Ahn & Kim, 2017*). Farmers may also be reluctant to apply BMPs due to economic reasons. Therefore, an integrated knowledge about farm characteristics and the environmental attitudes of farmers is required before adopting BMPs (*Liu & Brouwer, 2022*). *Dai et al. (2018)* proposed a combined model to create a series of BMPs placement schemes based on nutrients reduction and related costs. They concluded that nutrient load discharged into the lake and tributaries could be decreased to an acceptable level with a proper tradeoff between costs and risks (*Dai et al., 2018*). In a brief, recent studies imply that pollution reduction, applicability, and economic issues are the main concerns in BMP assessment. Nonetheless, their probable impacts on larger ecosystems and nutrition production require further evaluation.

Most of the literature has shown that the soil and water assessment tool (SWAT) was the main technique for integrated basin modeling. By this tool, the direct impacts of BMPs can be evaluated in hydrological response units (HRUs) and receiving water bodies (*Jamshidi, Imani & Delavar, 2020*). However, this simulation tool cannot account both direct and indirect cumulative environmental impacts (CIAs). The question of which BMP has the least total impacts on the ecosystem and food production still remains. Life cycle assessment (LCA) has the potential of answering this question through a data inventory that quantifies main ecological indices. These indices can translate data into ecological damages. It provides a framework for comparing strategies quantitatively based on their CIAs. For example, *Xu et al. (2017)* compared the CIAs of different low impact development BMPs as treatment systems (*Xu et al., 2017*). Comparing the sludge-dredging methods in Baiyangdian Lake, northern China (*Zhou et al., 2021*), treatment systems for Yangtze River rehabilitation (*Yao et al., 2021*), and sea water desalination (*Mannan et al., 2019*) are other applications of LCA in water quality management. Eutrophication is also a critical subject among the midpoint indices in life cycle impact assessment (LCIA) (*Cosme & Hauschild, 2017*; *Rosenbaum et al., 2017*). TN and TP concentrations in water directly affect this problem (*Chapra, 2008*), while other features, such as water consumption, are also effective on freshwater ecosystems, aquatic habitat or eutrophication intensification (*Damiani et al., 2019*). Since it is difficult to evaluate the eutrophication potential of agricultural systems, a combined method is required for the CIA of nutrients release from farmlands (*Ortiz-Reyes & Anex, 2018*).

The main purpose of this study is to develop a combined method based on SWAT-LCIA to evaluate and compare the CIAs of BMPs in a basin. The developed framework also introduces a state-of-the-art index for quantifying the food environmental footprint (FEF). This approach accounts related environmental damages of nutrition production in a basin and develops environmental perspective in water-food nexus problems. For these purposes, a lake basin is used as the study area to verify the proposed methodology. Here, the SWAT outputs are the main inventory of related midpoint indices in ReCiPe, a developed LCIA method (*Huijbregts et al., 2016*). Health, eutrophication, water consumption, aquatic and terrestrial ecosystems are the affected environments. Their CIA is normalized afterwards and evaluated by endpoint indices as ecological and health damages in ReCiPe. In addition, this research considers WF as the driving index for water consumption in LCIA and uses two different methods in calculations for weighting indices.

# MATERIALS AND METHODS
## Methodology
This study follows a 4-step combined methodology. In the first two steps, data is gathered and a basin is simulated by the SWAT model with the perspective of water quality and quantity. Here, the effectiveness of different BMPs on exporting pollution loads (kg/ha), pollutants concentration in lake (mg/L), crop production yields (ton/ha), nutrition production (Kcal/yr), and water footprint (m$^3$) are evaluated. Thus, the modeling provides a quantitative framework for further environmental-food analysis in basin. In this study,

the first two steps, except the nutrition production, follows the previously developed SWAT model by *Jamshidi, Imani & Delavar (2020)*.

In the third step and to quantify the CIAs of BMPs, a combined method is developed to convert the modeling results into equivalent environmental damages. An excel-base LCIA method according to ReCiPe (2016 v1.1) is used including related characterization midpoints (water consumption and eutrophication) and endpoints (human health and ecosystem damages) with normalization coefficients. In this step, some new approaches are considered to develop LCIA analysis. For example, the embodied water consumption, directly analyzed by the SWAT model (WF), is introduced as a reliable water consumption index for the LCIA of food crops. This is due to the fact that food crop's WF includes both consumed (blue and green) and polluted (grey) water. These items fit to life cycle assessment of available water in the ecosystem (*Bigdeli Nalbandan et al., 2022*).
In addition, this step considered two different weighting approaches for integrating health and ecosystem damages (endpoints) as a single index. The entropy analysis uses a mathematical equation to calculate the weights of health and ecosystem, while environmental performance index (EPI) applies predefined weights for the two endpoints.

In final step, a state-of-the-art index is introduced as "environmental footprint of food production" (FEF) that calculates the cumulative environmental damages of nutrition production in basins. This new index is applicable for quantifying the equivalent environmental damages related to food production. It also compares the cumulative impacts of BMPs and farm management practices by considering different perspectives like WF, pollution emissions, crop nutrition, and ecosystem protection. The main contribution of this research is in its combined method, particularly the third and fourth steps. Here, an environment-food nexus analysis compares the cumulative impacts of BMPs in a basin. The methodology steps are illustrated in Fig. 1.

Zrebar Basin, western Iran, was chosen as the study location to verify the method. It doesn't mean this approach is developed for a specific basin. On the contrary, the SWAT-ReCiPe is applicable in any basin for comparing farm management strategies.

## Study area

The proposed methodology is verified in Zrebar Lake basin, western Iran. Zrebar basin encompasses 90 km$^2$ including 20 km$^2$ of irrigated and rain-fed farmlands (22%). Its lake meets eutrophication problem mainly due to the agricultural discharges, particularly irrigated farmlands (*Imani, Delavar & Niksokhan, 2019*). Main rain-fed (RF) crops in this area are wheat, barley, grape and peas with average nutrition values of 3,640, 3,540, 670 and 420 cal/kg, respectively. The irrigated crops include tomato, tobacco, alfalfa, apple with average nutrition values of 180, 0, 230, and 520 cal/kg, respectively in addition to irrigated wheat and barley. Figure 2 shows the dominant LUs in the study area with its geographical conditions.

## Simulation-calibration

In the proposed methodology, the SWAT model is used for basin simulation before accounting environmental damages and footprints of agricultural productions. This model

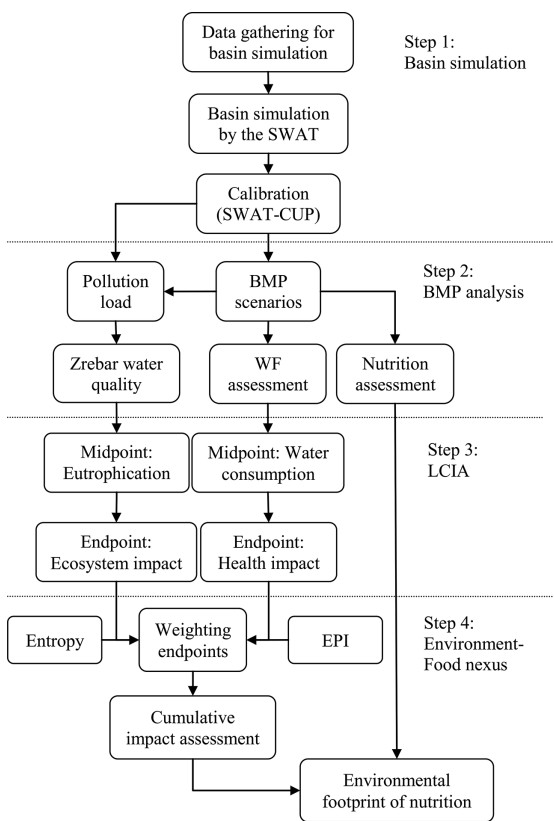

**Figure 1 Flow diagram of methodology and research steps.** Each box introduces the main activity in the proposed methodology and markers join boxes with their previous and next activities. The methodology is divided in four main steps.

can simulate complicated systems by considering management practices in farmlands, interactions between water quality and quantity, pollution transport, and production yields (*Abbaspour et al., 2015*; *Arnold et al., 2012b*; *Rivas-Tabares et al., 2019*). Therefore, required data such as topography, soil properties, LU type, management practices, and weather/climate were inputted to the model (Table 1). The basin was split into 26 sub-basins and 1,100 HRUs. This model was calibrated and validated based on available data (2006–2013) of monthly lake inflow, nitrate and phosphate concentrations simultaneously. Production yields and evapotranspiration rates were also controlled with the observation data (*Jamshidi, Imani & Delavar, 2020*). Table 2 outlines the calculated regression coefficient ($R^2$) and RMSE-observations standard deviation ratio (RSR) in the calibrated model.

It is noteworthy that the main idea of this research is to develop an integrated method for accounting environment-food nexus. Accordingly, authors used the outputs of the already calibrated SWAT model previously developed for BMP and WF assessment in the study area (*Jamshidi, Imani & Delavar, 2020*). Thus, simulation-calibration details are skipped here as they can be fully retrieved in the cited reference.

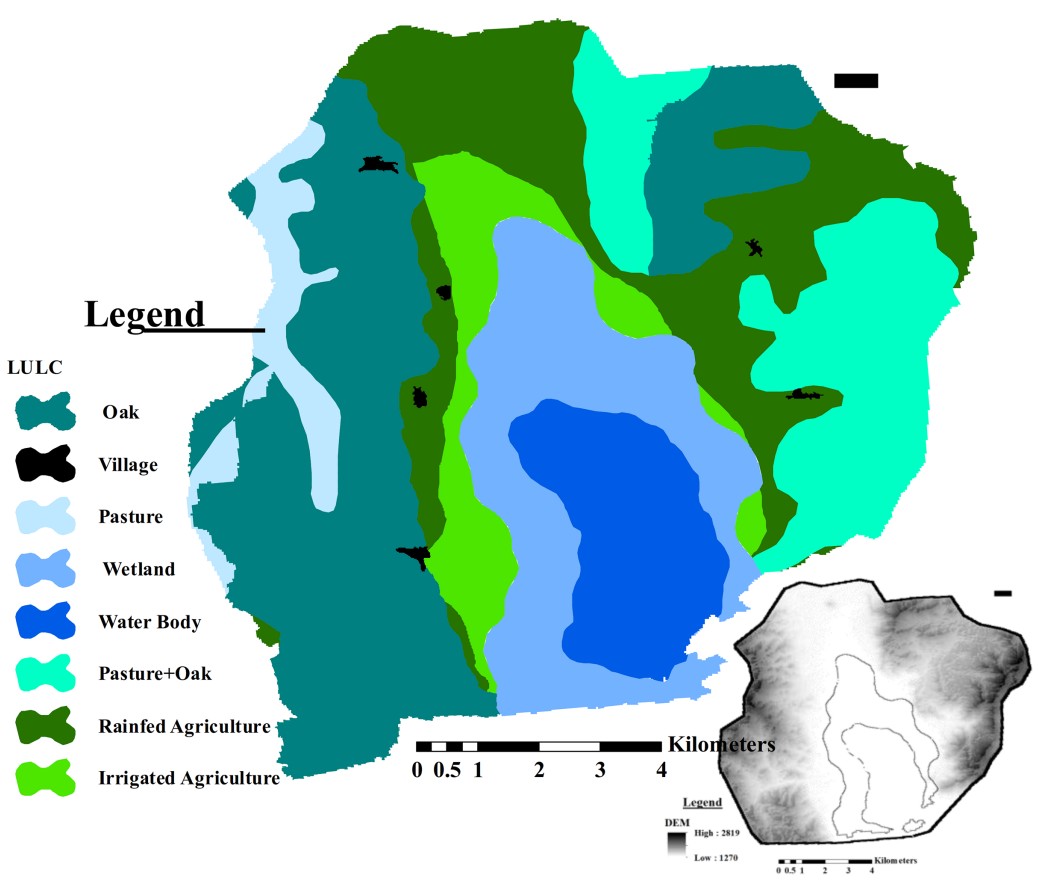

**Figure 2  Geographical status of Zrebar Lake basin with its land-uses.** This figure shows the DEM with main land-uses of the study area.                               

## BMP scenario

This study uses the SWAT outcomes for BMP analysis in three scenarios as defined in Table 3. Base is the scenario without using any BMPs. In BMP1, the application of fertilizers, manure and chemical, and water for irrigation are reduced 25% for farmlands. In BMP2, the reduction equals 50%. In both BMP scenarios, FS is assumed to be implemented in the vicinity of lake. Slim FS represents 10–12 m width, while moderate FS has 20–25 m width. All scenarios are analyzed by the SWAT model in the same period from 2007–2013.

## Water footprint

The WFs of agricultural productions are accounted by the standard method and include the three main elements of green, blue and grey water (*Franke, Boyacioglu & Hoekstra, 2013*; *Hoekstra et al., 2011*). It should be noted that WFs calculate the direct embedded water of farmlands and exclude indirect water embodied in further processing of agricultural productions.

**Table 1 Required datasets of SWAT and their sources.**

| Theme | Data basis | Source and scale |
|---|---|---|
| Topography | Digital Elevation Model (DEM) | National Aeronautics and Space Administration (NASA) database of the United States, 30 m resolution grid |
| Land use data | Land-use maps | Iranian Forests and Farms Organization (2006); 1,000 m resolution grid |
| Soil data | Soil properties, soil layers | Global soil map of Food and Agriculture Organization (FAO, 1995); 1:500,000 |
| Climatic data | Minimum and maximum daily temperature, mean monthly and daily precipitation, relative humidity, wind velocity | Iran Meteorological Organization (IRIMO) (2000–2014) |
| Hydrological data | Monthly flow rates, Lake water level, dam operation | Kurdistan regional Water Authority, Ministry of Energy (2006–2013) |
| Water quality | TN and TP concentrations of Lake | Department of Environment, Kurdistan province (2007 and 2010–2013), export coefficients for point sources (*Chapra, 2008*) |
| Management practices | Planting types, tillage, harvesting, grazing, fertilizers application and their types, irrigation | Ministry of Agriculture, Kurdistan province (2000–2014) |
| Crop yields | Typical yields and evapotranspiration data | Ministry of Agriculture, Kurdistan province (2014) |

**Table 2 Model performance in simulating the water quality and quantity of Lake Basin (*Jamshidi, Imani & Delavar, 2020*).**

| Parameter | Calibration | | Validation | |
|---|---|---|---|---|
| | $R^2$ | RSR | $R^2$ | RSR |
| Lake inflow (m³/s) | 0.64 | 0.41 | 0.76 | 0.22 |
| Nitrate (mg/L) | 0.89 | 0.62 | 0.70 | 0.70 |
| Phosphate (mg/L) | 0.64 | 0.34 | 0.30 | 0.38 |

**Table 3 BMP scenarios and their specifications.**

| BMP scenario | Management strategies |
|---|---|
| Base | Without BMP |
| BMP1 | 25% reduction of fertilizers and water for irrigation, with slim vegetated filter strip |
| BMP2 | 50% reduction of fertilizers and water for irrigation with moderate vegetated filter strip |

$$WF = GnWF + BWF + GWF \tag{1}$$

$$GnWF = 10ET_a \tag{2}$$

$$BWF = 10\left(ET_b - ET_a\right) \tag{3}$$

$$GWF = \max\left(\frac{L}{C_{max} - C_{nat}}\right)_i \tag{4}$$

**Table 4 Midpoint coefficients considering eutrophication in different environments.**

| Environment | Effective ecosystem | Midpoint conversion coefficients (M) | | | | equivalent unit |
|---|---|---|---|---|---|---|
| | | Nitrate | Nitrite | phosphorus | Phosphate | |
| Fresh water | Fresh water | - | - | 1 | 33 | kg P-eq. to freshwater/kg |
| | Marine | 0.07 | 0.09 | - | - | kg N-eq to marine water/kg |
| Marine water | Fresh water | - | - | | | kg P-eq. to freshwater/kg |
| | Marine | 0.23 | 0.3 | - | - | kg N-eq to marine water/kg |

In these equations, $GnWF$, $BWF$ and $GWF$ are green, blue and grey water footprints (m³), respectively. $ET_a$ refers to the evapotranspiration from soil and vegetations when there is no irrigation (mm). $ET_b$ includes the total evapotranspiration during irrigation ($ET_b > ET_a$). Thus, the SWAT models evapotranspiration twice with and without irrigation. It uses the climatic data of minimum and maximum daily temperature with precipitation as mentioned in Table 1. Afterwards, it estimates the actual evapotraspiration for crops by Hargreaves equation (*Kisi, 2007*; *Majidi et al., 2015*). $L$ is the exported pollution loads (ton/ha) of pollutant $i$ to the receiving water body. In modeling, output.hru file shows the pollution loads per each HRU and sub-basin. $L$ is the net pollution loads transported from LUs into the 26th sub-basin, the Zrebar Lake in this study (*Arnold et al., 2012a*). $C_{max}$ is the maximum allowable concentration of pollutants. $C_{nat}$ equals pollutants concentration in the receiving water on the condition there is no human interference. Here, the $C_{max}$ of TN and TP are assumed constant according to the global limits for controlling the trophic state of lakes (*Jamshidi, 2021*) and they equal 1.5 and 0.035 mg/L, respectively. $C_{nat}$ of TN and TP are also assumed 0.4 and 0.01 mg/L, respectively (*Jamshidi, Imani & Delavar, 2022*).

### Environmental impact assessment

The quantification method of environmental damages in basin is compatible with the indices of LCIA. In the current research, LCIA characterization coefficients are derived according to the ReCiPe method, which was previously developed by some collaborations in Europe (*Huijbregts et al., 2017*). In this method, normalized data at the European and global level are available for 16 midpoint and three endpoint indices. In later updates, ReCiPe considered several conversion coefficients based on global scale, instead of the European scale. However, it preserved the possibility of using these coefficients on the continental and country scales. Another feature of ReCiPe is to expand environmental damages for evaluating the impacts of water consumption on human health, aquatic and terrestrial ecosystems (*Huijbregts et al., 2017*). However, the current study proposes to use WF for accounting the water consumption of food crops in LCIA. This is due to the ability of WF in calculating water consumption including both water quality and quantity.

In this method, all effective environmental factors derived from the SWAT model are initially converted into the equivalent units. Table 4 illustrates eutrophication midpoint coefficients that convert $NO_3$, $NO_2$, $NH_3$ and $PO_4$ to the equivalent environmental damages. The average WFs of crops are considered (m³) for water consumption midpoint

**Table 5 Endpoint coefficients to convert midpoints into equivalent environmental damages.**

| Environment | Midpoint indicator | Endpoint conversion coefficient (E) | Equivalent unit | Normalization index (N) |
|---|---|---|---|---|
| Human health | Water consumption | 2.22E−06 | Daly/m$^3$ consumed | 1.96E−04 |
| Terrestrial ecosystems | Water consumption | 1.35E−08 | Species.yr/m$^3$ consumed | 3.48E−06 |
| Freshwater ecosystems | Eutrophication | 6.71E−07 | Species.yr/kg P to fresh water eq. | 4.90E−07 |
| | Water consumption | 6.04E−13 | Species.yr/m$^3$ consumed | 6.16E−10 |
| Marine ecosystems | Eutrophication | 1.70E−09 | Species.yr/kg N to marine water eq. | 6.12E−09 |

in aquatic, terrestrial and marine ecosystems. Equation (5) shows how conversions are carried out.

$$Q_j = (T \times M)_j \tag{5}$$

$Q$ is the midpoint index, $T$ represents the output of the SWAT model such as water footprint or pollutant concentration, $M$ is the conversion coefficients, and $j$ is environmental component such as aquatic, terrestrial, and marine. By this equation, it is possible to calculate the equivalent environmental effects of each pollutant in the life cycle period of the product or activity. It should be noted that these coefficients represent average values. It means that they do not need supplementary conversion coefficients for shallow or deep waters as they are free from adjustment for the conditions with different vegetation or trophic state. Moreover, pollution discharge to freshwater has indirect impacts on other ecosystems in long-term. Thus, marine impacts are also considered in calculation even the pollution is not directly discharged to the sea.

Since the midpoint indices are calculated based on equivalent units, such as kgN-eq or m$^3$ water consumed, it is necessary to accumulate these environmental impacts with different units under a single index. This is the most challenging step in conventional CIA. ReCiPe uses equivalent damage-based indices for integrating midpoints into endpoints by Eq. (6).

$$D_j = (Q \times E)_j \tag{6}$$

Here, the calculated midpoint indices ($Q$) are converted into endpoint damage-based indices ($D$) according to conversion coefficient of $E$ (see Table 5). Here, human health and ecosystem (non-human) damages are two endpoint indices. The former is based on disability-adjusted life years (DALY) and the latter is based on probable number of harmed species in year (species.yr). DALY represents the equivalent years of human life lost by death or being disabled due to illness caused by existing pollutants in the environment. On the other hand, the unit of measuring ecosystem damage is the total number of species lost over time. Table 5 shows the conversion coefficients that turn each equivalent midpoint indices into the two endpoints. ReCiPe model also recommends that endpoints ($D$) should be normalized by specific coefficients that turn the calculated damages into dimensionless indices per person (*Sleeswijk et al., 2008*).

## Normalization and weighting

Calculated endpoints are normalized by Eq. (7) on a global scale based on reference coefficients (Table 5). They are finally aggregated according to their weights by Eq. (8). In this study, entropy and EPI are the weighting methods of normalized endpoints.

$$R = \frac{D}{N} \qquad (7)$$

$$C = \sum (W \times R) \qquad (8)$$

where, $C$ is the annual environmental damage per person, $W$ is the weight of each endpoint, $N$ represents the normalization value and $R$ is the normalized endpoint. Weights can be calculated based on different mathematical methods, such as entropy or fuzzy (Chen et al., 2019; Zeng, Luo & Yan, 2022), or based on expert opinions and references (Chen et al., 2022). In this study, EPI determines health and ecosystem weights as 0.4 and 0.6, respectively (Hsu & Zomer, 2016), whereas entropy method ($W_{En.}$) calculates the weights of endpoints through a probabilistic function as Eq. (9).

$$W_{En.} = -\frac{1}{\ln(t)} \sum_{z=1}^{t} (R \times \ln R)_z \qquad (9)$$

In which, $t$ is the number of available data. In entropy, factors with more data dispersion gain higher weights (Imani, Delavar & Niksokhan, 2019). Here, the weights of endpoints ($R$) are evaluated based on $C$ variations from 2007–2013 in each BMP scenario. Accordingly, the ecosystem and health endpoints weigh 0.44 and 0.56, respectively in the entropy method.

## Environmental-food index

This study introduces a new index for food and nutrition production in farmlands. It is quantified based on the environmental damages calculated by the SWAT-ReCiPe. This index quantifies the CIA per food production in any area or BMP as Eq. (10).

$$FEF = \frac{C}{S} \qquad (10)$$

In this equation, $FEF$ is a dimensionless index that represents the CIA of food production. In other words, $FEF$ is the environmental footprint of nutrition production. It can be calculated by the proposed method for comparing major environmental concerns in food production, including water-food nexus. Low $FEF$ (~0) means that strategies used for food production is rather clean, while higher $FEF$ (>1) indicates their destructive condition. $C$ is defined earlier that notes environmental damages (CIA), and $S$ is calculated by Eq. (11).

$$S = \frac{T_{Cal}}{B \times P} \qquad (11)$$

In which $T_{Cal}$ is the daily total nutrition (calories) of food production in the study area, $B$ equals the malnutrition baseline of humans assumed 2,000 cal/day (Liu et al., 2022), and

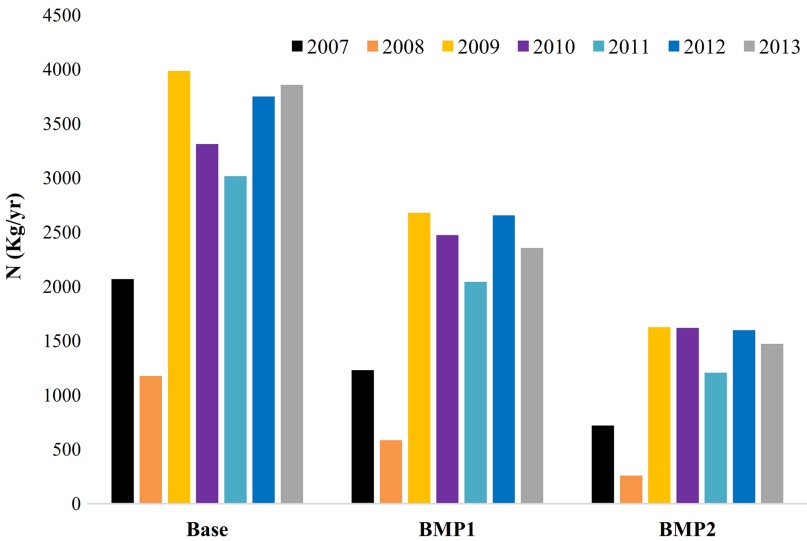

**Figure 3 Accumulated annual N pollution exported by farmlands in management scenarios (2007–2013).** The total nitrogen pollution loads discharged to the lake from all HRUs. Each column represents a year of simulation for BMP scenarios.

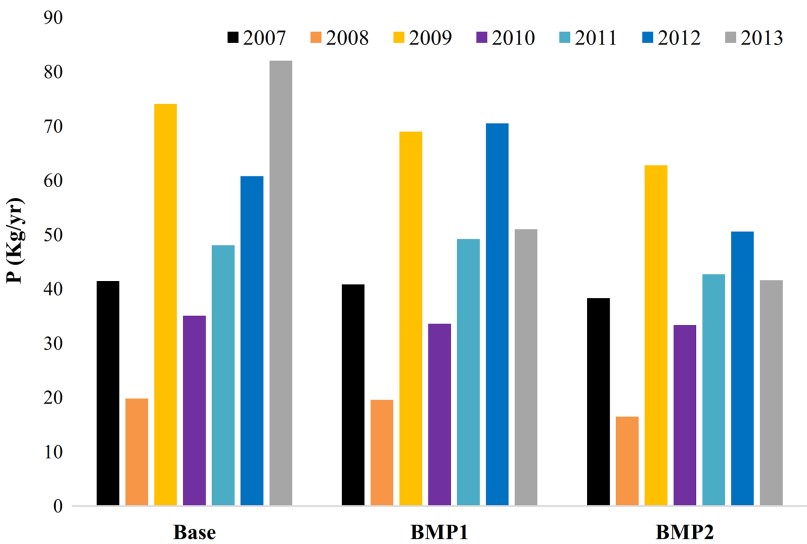

**Figure 4 Accumulated annual P pollution exported by farmlands in management scenarios (2007–2013).** The total phosphorous pollution loads discharged to the lake from all HRUs. Each column represents a year of simulation for BMP scenarios.

$P$ is the global population (7.75 billion) to convert and normalize $S$ per person in global scale.

## RESULTS AND DISCUSSIONS

### SWAT outcomes

The basin simulation by the SWAT model could calculate the annual pollution loads exported by HRUs in different management scenarios. Figures 3 and 4 respectively show the

**Table 6 Outputs of the SWAT model in different BMP scenarios.**

| LU | Base | | | BMP1 | | | BMP2 | | |
|---|---|---|---|---|---|---|---|---|---|
| | Yield (ton/ha) | WF (m³/ton) | Nutrition (MCal/yr) | Yield (ton/ha) | WF (m³/ton) | Nutrition (MCal/yr) | Yield (ton/ha) | WF (m³/ton) | Nutrition (MCal/yr) |
| Alfalfa | 4.5 | 2,699.7 | 91 | 3.93 | 2,673.5 | 79 | 3.4 | 2,529.3 | 68 |
| Apple | 11.2 | 1,065.8 | 431 | 9.28 | 1,114.1 | 357 | 7.4 | 1,144.9 | 284 |
| RF barley | 0.9 | 3,521.0 | 464 | 0.90 | 3,499.3 | 464 | 0.9 | 3,451.3 | 464 |
| Barley | 2.1 | 2,095.1 | 200 | 2.10 | 2,048.8 | 200 | 2.1 | 1,953.1 | 200 |
| RF Pea | 0.5 | 8,497.6 | 61 | 0.50 | 8,479.7 | 61 | 0.5 | 8,477.3 | 61 |
| RF grape | 4.0 | 939.5 | 536 | 4.00 | 936.1 | 536 | 4.0 | 927.7 | 536 |
| Tobacco | 1.9 | 4,149.6 | 0 | 1.67 | 4,221.0 | 0 | 1.5 | 4,075.8 | 0 |
| Tomato | 11.1 | 974.3 | 27 | 9.10 | 1,028.1 | 22 | 3.7 | 2,077.4 | 9 |
| RF wheat | 1.1 | 2,926.1 | 4,004 | 1.10 | 2,909.0 | 4,004 | 1.1 | 2,872.6 | 4,004 |
| Wheat | 2.8 | 2,206.7 | 1,303 | 2.74 | 2,196.1 | 1,275 | 2.6 | 2,173.0 | 1224 |

cumulative N and P loads discharged by farmlands in three scenarios. Here, the annual variations are due to (1) the precipitation variation influencing pollution transport from RF farms, and (2) temporary water transfers from upstream for irrigation development. The average N pollution of all irrigated and RF farms ranges between 1,176 and 3,985 kg yr$^{-1}$. This value for $P$ is between 20 and 82 kg yr$^{-1}$. For 20 km$^2$ farming area, the average export coefficients for N and P are 0.6–2 kg ha$^{-1}$ yr$^{-1}$ and 0.01–0.04 kg ha$^{-1}$ yr$^{-1}$, respectively. This range implies that nutrient export coefficients can increase 3–4 times greater than dry periods in the study area. On an average for 2007 to 2013, BMP1 can reduce 33.8%N and 7.7%P pollution exported from all agricultural LUs. BMP2 can improve these removals to 59.9% and 20.9% for N and P, respectively. It implies that basin response to BMPs' is not linear. In addition, P removal requires stricter BMPs than N removal. Yet, nutrient pollution reduction may have different ecological impacts on marine, aquatic and terrestrial systems which are accounted through the combined method. BMPs are also effective on crops production yields, WFs and nutrition productions (Table 6).

### Environmental impacts

For the base scenario, the combined method calculates the environmental midpoint impact ($Q$) of farming in Zrebar basin as Fig. 5. It shows that freshwater eutrophication is the most critical item during the study period. The embodied water consumed is also significant for damaging the terrestrial ecosystem and human health. This conclusion remains unchanged in BMP1 and BMP2 despite 25–50% FR (Fig. 6). Since TP concentration in lake is the main driver of freshwater eutrophication, consuming less ammonium-based fertilizer can hardly solve eutrophication problem in short term in the combined method. On the contrary, controlling erosion and sediment transport by FS from upstream is more efficient for TP mitigation in the lake.

Figure 7 shows the cumulative ecological damages. Their values in Zrebar basin are relatively larger than health problems in all management scenarios. It is noteworthy that human health indices are mostly reliant on toxins and heavy metals. These pollutants were

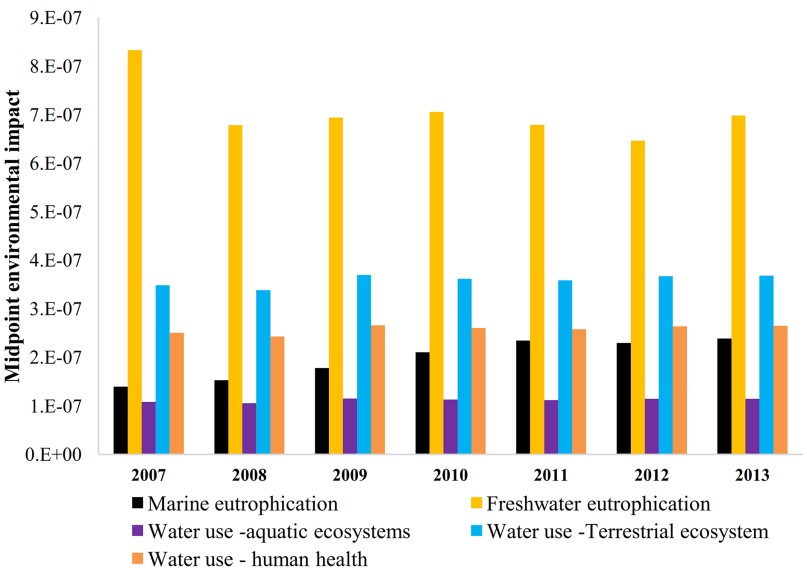

**Figure 5 Environmental impacts of farming based on midpoints without using BMPs (base scenario).** In the base scenario, the five main midpoints of this study are calculated by the SWAT model seperately for each simulation year (2007–2013).

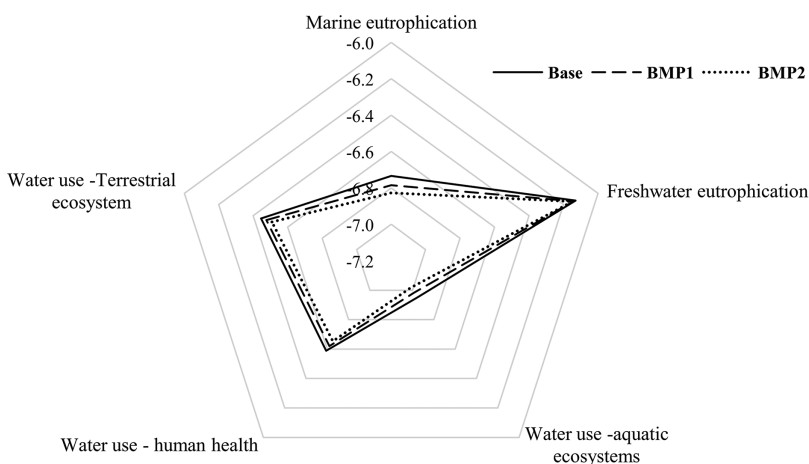

**Figure 6 The environmental impacts of management practices based on five midpoints on average for study period (2007–2013).** The sharpest corner of diagram points to the highest midpoint (impact) in BMP scenarios. Data are based on average results of 2007–2013.

hardly traced in the study area. The results present that farm management strategies can mitigate the average ecological impacts from 1.41E−6 to 1.34E−6 (4.9%) for BMP1 and 1.28E−6 (9.2%) for BMP2. Likewise, these strategies can diminish human health risk from 2.58E−7 to 2.4E−7 (6.8%) for BMP1 and 2.22E−7 (13.9%) for BMP2. It means that using 50% less fertilizer with a FS in this area may totally reduce 9% ecological and 14% health risks (Fig. 8). Here, the cumulative impacts are low but not negligible as they range between 1E−6 and 1E−7 per person. However, these values are meaningless unless they are used as quantitative tools for comparative analysis.

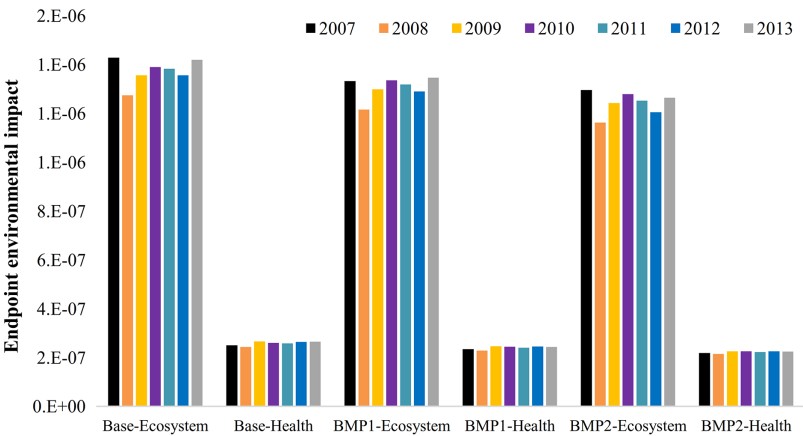

**Figure 7 The annual endpoint environmental damages of management practices.** Calculated endpoint indices of ecosystem and health per BMP scenarios and simulation years (2007–2013)

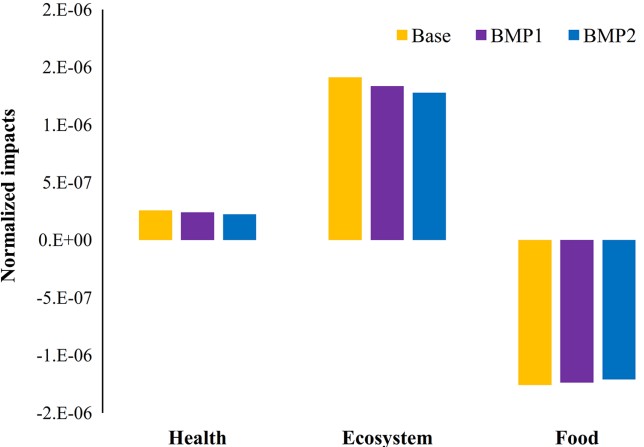

**Figure 8 The impacts of management practices on endpoints and food production on average for study period (2007–2013).** Agricultural productions in this area have adverse impacts on ecosystem and health, while have constructive impacts on food and nutrition values. BMPs reduce adverse impacts but lose some nutrition production.

Figure 8 summarizes the impacts of BMPs on food production ($S$) in addition to normalized environmental impacts (per person) on the ecosystem and health. Since nutrition production is intrinsically a favorable action with environmental perspective, their related impacts are negative. The overall environmental impact of farming and related management practices are finally calculated by the weighted average of normalized ecosystem and health damages. The weighing step is carried out with different methods. Since EPI gives higher weights to ecological items, the related results are relatively more than entropy method. Despite different weighting approaches, the overall CIA ($C$) reduction for BMP1 ranges between 5–8%, while it ranges between 10–13% for BMP2. It implies that using strict BMPs may not necessarily have significant improvement. On the contrary, $S$ is reduced 1.66% and 3.73% by BMP1 and BMP2, respectively. It points an

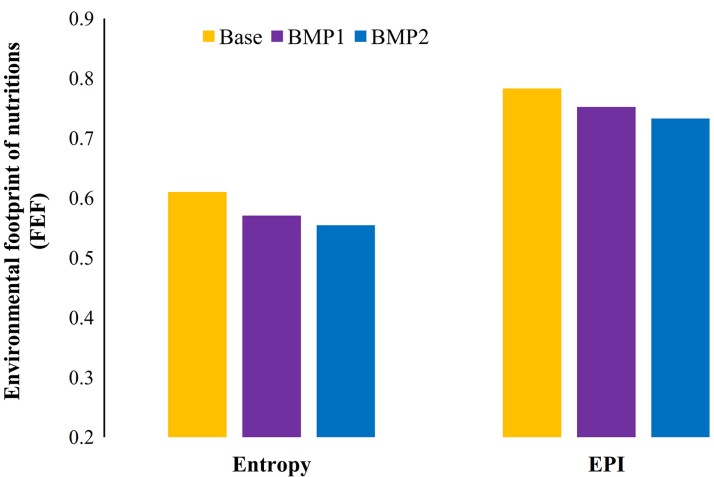

**Figure 9 Average environmental footprint of food production in different BMPs and weighting methods for study period (2007–2013).** The overall negative and positive impacts of agricultural productions on environment and food production is combined within FEF. Two weighting methods present different values for this footprint.

important conclusion that although farm management practices may reduce environmental damages, they can adversely reduce the nutrition production. This conclusion highlights an environment-food nexus index for more comprehensive understanding of management impacts.

Figure 9 draws the environmental footprint of nutrition production (FEF) in Zrebar Basin. Here, the conventional pattern (base scenario) can quantitatively generate 0.61 (entropy) and 0.78 (EPI) environmental impacts. In other words, 0.61–0.78 environmental units would be damaged for one unit nutrition production. This is a footprint mainly accounted by the impacts of consumed water and eutrophication in the study area originated by the agriculture. Using BMP1 and BMP2 can reduce FEF 6.5–9.1% (entropy) and 4–6.4% (EPI), respectively. It means that 50% FR combined with FS (BMP2) can reduce 6.4–9.1% of FEF in Zrebar basin. Obviously, this new index is more helping for policy makers rather than conventional approach on pollution reductions in a basin. For example, this index can present criteria to compare two alternatives of implementing vegetated FS or changing crop patterns in a basin. The first alternative only reduces pollution loads and consequently environmental impacts, while the second option emphasizes nutrition improvement despite pollution discharges.

## DISCUSSION

What stands out this research and makes it different with previous literature is combining SWAT-ReCiPe for accounting the new damage-based index of FEF. This idea provided a quantitative solution to include water quality issues within water-energy-food nexus problems (*Heal et al., 2021*). It is verified in a lake basin with different irrigated and rain-fed farming.

Results showed that N and P pollution removal by BMPs in Zrebar Lake basin varies between 34–60% and 8–21%, respectively. These ranges are comparable with recent

literature. For example, *Venishetty & Parajuli (2022)* estimated 25% N and 10% P removal by FS and the riparian buffer zone (*Venishetty & Parajuli, 2022*). In another study, the minimum impact of FS on TN and TP removal was 29% and 42%, respectively (*Risal & Parajuli, 2022*). In a river basin with 20,000 km$^2$ area, P export coefficient was 0.63 kgha$^{-1}$yr$^{-1}$ and FS could reduce 4–20% of its pollution (*Almendinger & Ulrich, 2017*). *Ricci et al. (2022)* also estimated TN and TP export coefficients about 49 kg ha$^{-1}$y$^{-1}$ and 0.044 kg ha$^{-1}$y$^{-1}$, respectively in a river basin. Here, BMPs could remove 20% nutrient pollution loads from farmlands (*Ricci et al., 2022*).

The current study also implied that BMPs may have secondary impacts due to long-term terrestrial and aquatic pollution transport, water consumption, or changing LUs. Similar conclusion has been recently achieved by *McAuliffe, Zhang & Collins (2022)*. They highlighted that direct short-term water quality rehabilitation, such as TN and TP reduction, may not necessarily ends into a sustainable strategy. With the perspective of integrated environmental management, on-farm intervention strategies have by-effects that should be considered in decision-making (*McAuliffe, Zhang & Collins, 2022*). The proposed method can more or less consider these impacts *via* LCIA. However, midpoint indices can be different on the subject of basin specifications. For example, water consumption and eutrophication are the main environmental issues in the current study. In different regions, other environmental issues like global warming, LU change, and even air pollution have related indices in ReCiPe (*Huijbregts et al., 2017*). Variety in midpoint indices may not limit SWAT-ReCiPe and FEF application. On the contrary, its multidisciplinary specification develops its purpose to calculate broader range of environmental damages for integrated monitoring and problem solving. For example, it is conventionally believed that hydropower systems in water reservoirs are a renewable energy source and environmental friendly. Nevertheless, it is recently noted that these systems can be the significant sources of greenhouse gas emissions due to their long-term secondary limnology and ecological impacts (*Gemechu & Kumar, 2022*). *Čuček, Klemeš & Kravanja (2015)* recommended LCA method for environmental assessment because of the chance of using footprints, such as carbon footprint, biodiversity footprint, ecological footprint, *etc.* (*Čuček, Klemeš & Kravanja, 2015*). These indices can help to account cumulative environmental footprint of productions within LCIA similar to the method used for WF in this study. It is noteworthy that recent literature has also focused on developing social LCA indices (*Bouillass, Blanc & Perez-Lopez, 2021*; *Siebert et al., 2018*). This perspective aims on integrating social with environmental-based LCA indices. In other words, the safe and healthy living conditions of farmers, their employment, social fairness, and public commitment to sustainability would also be important in decision-making (*Kühnen & Hahn, 2017*). Thus, we recommend future studies to assess the cumulative impacts of BMPs based on the combined social-environmental indices of LCIA before FEF evaluation.

This study also applied the SWAT model for basin simulation. It could present a reliable framework for integrated LCIA, WF and water quality assessment in different BMPs. It is noteworthy that using WF is more efficient in LCIA than typical water consumption. It has two reasons: (1) WF is the embodied water in production. Thus, it is compatible with other

LCA indices as both consider indirect impacts, (2) WF includes equivalent water pollutions in form of GWF within the consumed water. GWF is an exceptional index for LCIA as it bridges water pollution to unavailable water for health or ecosystem consumption. It means that GWF is the only functional index that enables LCIA to include the indirect impacts of water pollution on destroying water resources. Recent studies could even develop the understanding about GWF. In new definition, regional ecological impairments are decisive for GWF calculations (*Jamshidi, Imani & Delavar, 2022*). The interactions between water resource and ecological assessment does not limit to WF assessment. A recent study used environmental Kuznets curves with the SWAT model. Researchers assessed the relationship between environmental degradation and developing agriculture (*Golzari et al., 2022*). However, the current study could develop a quantifiable method for integrated water and environmental assessment based on WF, LCIA and FEF indices. The proposed method can also be applicable for climate change. *Delavar et al. (2022)* showed the applicability of the SWAT model for water accounting during both wet and dry periods of climate change (*Delavar et al., 2022*). On the other hand, the effectiveness of 171 BMPs on reducing TN and TP were previously analyzed by LCA approach (*Chiang et al., 2012*). Despite the abilities of the SWAT model for basin water quality modeling (*Bigdeli Nalbandan et al., 2022*; *Li et al., 2021a*), this method has limits on accurate simulation of some pollutants. Toxins, heavy metals, and microbial pollution require accurate simulation as their impacts on health and ecological midpoint indices are critical in LCIA. However, they are sensitive to contamination transports and environmental conditions. Erosion, sediment adsorption and re-suspension, biomass accumulation, and volatilization are different transports that increase the uncertainties of both field samples and simulated results (*Du, Shrestha & Wang, 2019*; *Ouyang et al., 2018*). Further studies can focus on finding proper tools for simulating these pollutants in combination with LCIA. Since uncertainty is the main drawback of the proposed method, authors recommend it as an applicable tool for comparing the effectiveness of different strategies respecting their CIAs and FEFs.

## CONCLUSIONS

Pollution control is only one pillar of BMPs' sustainability assessment. Their impacts in larger ecosystems are also crucial for integrated decision-making. This study developed a method that combines basin simulation and life cycle impact assessment (LCIA). The soil and water assessment tool (SWAT) simulates the basin, while ReCiPe uses the modeling results as an inventory for LCIA. This approach has some advantages for the sustainability assessment of BMPs:

- It is a quantitative tool based on various environmental indices. The cumulative environmental impact accounts possible aquatic, terrestrial and marine impairments. Thus, it can simplify integrated evaluations and comparing BMPs.
- It is flexible to include new or integrated indices. The food environmental footprint (FEF) is a state of the art index that quantifies the total environmental damages of one

unit nutrition production. In a nutshell, FEF can add environmental footprint in water-food-energy nexus problems.

In the study area, fertilizer reduction and filter strip were effective on controlling nutrient pollution without notable negative impacts. BMPs reduced FEF and the water footprint (WF) and improved eutrophication problem. However, uncertainties were the main limits and drawbacks. These uncertainties are mainly reliant on LCIA coefficients and modeling pollution transports. Thus, this idea is recommended as a tool for comparing strategies instead of reporting certain results. Future studies can focus on upgrading this method. Developing indices, variable midpoints and footprints, besides social indices are some possible research areas.

### Funding
The authors received no funding for this work.

### Competing Interests
The authors declare that they have no competing interests.

### Author Contributions
- Shervin Jamshidi conceived and designed the experiments, analyzed the data, prepared figures and/or tables, authored or reviewed drafts of the article, concenptualization, supervision, administration, and approved the final draft.
- Anahita Naderi performed the experiments, analyzed the data, prepared figures and/or tables, and approved the final draft.

### Data Availability
  The raw data is available in the Supplemental Files.

### Supplemental Information
Supplemental information for this article can be found online at http://dx.doi.org/10.7717/peerj.14816#supplemental-information.

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
