# Peer review of "A quantitative approach on environment-food nexus: integrated modeling and indices for cumulative impact assessment of farm management practices"

_PeerJ, doi:10.7717/peerj.14816_

## Round 0.1 · original submission · Major Revisions

Dear Authors,

As you shall see that the reviewers have now commented on your manuscript. They are suggesting significant revisions in your manuscript. The major highlights of the revision are weak results and discussion section, an incomplete description of the methodology, along some bias in all the sections of the manuscript. I would request you kindly go through the suggested comments and revise your manuscript accordingly. Please do not forget to consider the revisions by reviewer 3 in the attached annotated manuscript file. I shall be looking forward to receiving your manuscript on time.

Best Regards
Gowhar Meraj

·

Basic reporting

This paper is generally fine while needing some minor revision.

Experimental design

Method is described in details.

Validity of the findings

Results and discussion should be more elaborated.

Additional comments

The authors have tried to establish a combined methodology with simulation basin by the SWAT model and LCIA by ReCiPe, and they also use the method to quantify food environmental footprint. This topic is of significance in farm management practices. Overall, this paper is well written and method is sufficiently described. I would like to recommend accepting it after addressing some minor issues.

——The literature review in introduction has potential to improve its logic, and some contents should be more concise. I know the paper in Peer J has word limitation, so please make some spaces for results and discussion, which are now very weak.
——I know the method is sufficient enough, but it is too big. Please consider condensing it.
——Please link your results to the discussion on farm (or other similar fields) management practice.
——As I mentioned before, the paper lacks insightful discussion, which I can not agree. Please do this.
——Limitation should be mentioned.
——I have a major concern, in your method, you solely combined some well-developed and used model (SWAP, LCIA). It appears that the novelty is not strong. Please highlight your key contributions.
——There are some minor language problems, please check and make corrections.
——It is unnecessary to abbreviate all terms especially for those not used often.

Reviewer 2 ·

Basic reporting

• The Literature references account 44, which is sufficient to provide the background. In addition, the introduction was appropriate and well-referenced. Also,13 references are from 2020, and four are from the corresponding author.
• The English language is understandable, but there are too many passive voices, long paragraphs, unfitting prepositions, repetitive words, grammar gaps, and missing articles and periods, among other faults. A proficient English-speaking editor should correct the manuscript before considering its acceptance.
• The manuscript structure is in the suggested format of standard sections, including the authors' contributions. Tables have no headings and notes, and they should have. No captions for Figures? The resolution should be improved.
• Raw data is available after the Data Sharing policy.
• The submission is ‘self-contained,’ as it includes relevant results related to the hypothesis.

Experimental design

• The manuscript comprises original primary research and fits the “Aims and Scope” of the journal on the subject “Environmental Sciences.”
• The objective is clear and fills a knowledge gap because the proposed Index is relevant and a novelty in measuring sustainability in agricultural food production and the environment.
• It described a rigorous investigation with appropriate technical & ethical standards, and the entropy method is practical for weighting and normalizing the index design.
• The methods are described with sufficient detail and information to replicate by researchers. Indeed, the methodology is recommended to apply in any basin, and, in my opinion, it will be helpful for the scenario´s analysis.
• The section does not mention the method selected to calculate the evapotranspiration in SWAT. And for the Water Footprint, how was it calculated? Wich Temperature? Were they calculated by the same method?
• The base scenario is the one with no BMP and the other with two positive perturbations, OK. But what about the time? Line 281: “Since this study evaluates C from 2007-2013 for each BMP…” The time is not clear. One scenario for the period?

Validity of the findings

• The proposed index and the methodology might be interesting to researchers on subjects as water, environmental, nutritional, agricultural, and food sustainability. The results are not derived from existing work and include software validation and verification with mathematical tools: entropy analysis and environmental performance index.
• The novelty and discussion are well done, but it can be improved with the statements of the conceptual model, boundaries, and limitations.
• The data is robust and statistically validated.
• The conclusions are appropriately stated, and connected to the innovative objective. Please, include the limitations of the methodology.

Additional comments

The manuscript has valuable results. The proposal is attractive and suitable to achieve quantitative criteria for sustainability in the environmental-food nexus, a new-fangled subject. The local approach, such as farms or basins, is the best for diagnoses based on indices and indexes and for reaching real solutions. The Index is a novelty, and it was well stated.

Nevertheless, the manuscript should be complemented. In addition to the English language, my principal concerns are the following:

-Evapotranspiration is usually responsible for errors in water calculation results. Please, include which method was selected in SWAT and details about the Water Footprint.
-The scenarios are well considered, but the year of the base scenario (without BMP) and the others should be explicit in the text, tables, and figures captions. Is the mean value of 2007-2013?
-Please state the year of base and scenarios, not only the geographic details and data of the case study. Are they all in the same year?
-About the previous concern, I suggest defining better the statements of the conceptual model before the performance of the mathematical one. As it turns out, for a general methodology, the boundaries should be more specific, as well as the limitations. The latter should be mentioned in the conclusions as they are the most important criteria to apply in other places.
-I did not find the figure captions and table headings. I hope they are somewhere. Please, improve figure resolutions.
-Finally, a question. Could the authors discuss deeply how important is or is not the Grey Water footprint?
Minor:
I am unsure about the figures´ resolution because I did not find the dimensions or the weight in the instructions for the authors. I noticed that they are out of the current standards of other journals, except for Figure 2. Better improve them.
Please write the whole words in the Conclusions, not the acronyms. Some acronyms are not detailed (ReCiPe).
Indicator, indice, and index? Please, check when using those words throughout the manuscript. Some authors do not agree with “indicator” and recommend “indice” instead.
Why is eutrophication always written with a capital letter?

·

Basic reporting

This is very interesting study and relevant for policy makers. The methodology to analyze the nexus is good.

Experimental design

Although the methodology is good, but for the first section i.e. pollution load calculation, author need to add the equation they have used. Also insert a table to report input data you have used and their sources.

Validity of the findings

No comments

Additional comments

Result and discussion is very weak. This need to be strengthen before this study can be accepted.
Please see the reviewed file for more comments.

---

## Round 0.2 · Minor Revisions

Dear Authors,

Thank you very much for revising your manuscript and considering the reviewers' comments. Please reduce the length of the abstract. Further, there are language issues that must be taken care of.

While submitting the revised manuscript for the second round of review, please accept the changes from the first round and only show the tracked changes for the second round of review.

I am looking forward to receiving your manuscript at the earliest.

Best regards
Gowhar Meraj

·

Basic reporting

The revised manuscript is well-prepared and my comments in the last review are adequately addressed. I merely have one additional suggestion. Please reduce the abstract. Now, it is very large. Also, some minor language issues exist, please correct them.

Experimental design

No comments

Validity of the findings

No comments

Additional comments

No comments

·

Basic reporting

As authors have addressed all my concerns and I don't have any additional comments, I vote for its acceptance for the publication.

Experimental design

No comment

Validity of the findings

No comment

---

## Round 0.3 · accepted · Accept

Thanks for taking care of all the suggested changes. I congratulate you for this very good work.